# Gender bias in the Chilean public health system: Do we all wait the same?

Susana Mondschein[1,2]*, Maria Quinteros[3,4], Natalia Yankovic[5]

**1** Industrial Engineering Department, Universidad de Chile, Santiago, Chile, **2** Instituto Sistemas Complejos de Ingeniería, Santiago, Chile, **3** Facultad de Ingeniería y Ciencias, Universidad Adolfo Ibañez, Santiago, Chile, **4** Universidad Finis Terrae, Santiago, Chile, **5** ESE Business School, Universidad de los Andes, Santiago, Chile

* susana.mondschein@uchile.cl

**Data Availability Statement:** All data is available from: Susana Mondschein, Maria J. Quinteros, Natalia Yankovic, 2020, "Replication Data for: Gender Bias in the Chilean public health system: Do

## Abstract

### Background

In 2002, Chile introduced a major health reform, designed to level out inequities in health-care coverage, access and opportunities. In particular, the opportunity guarantees ensure a maximum time to receive the appropriate diagnosis and treatment, and thus, gender bias should not be observed.

### Objective

To explore the existence of differences in the timeliness of treatment between women and men under the Chilean public health insurance system. We controlled by other observable variables, including age, insurance holder status, provider complexity and health district.

### Methods

We used an individual level database that includes all interactions for the diseases covered under the national plan from 2014 to 2019. We excluded from the analysis the diseases affecting only men, women, and infants. To study the waiting time differences between women and men, we first perform a Welch two sample t-test. Then, we used a multilevel hierarchical regression model to further explore the impact of gender in waiting time. At the individual level, we included gender, insurance holder status, age, and the interaction between gender and age. For the aggregate levels, we used the specific opportunity guarantee, the type of provider, and health district.

### Results

From the Welch two sample t-test, we found significant differences in waiting times between women and men, in seven opportunity guarantees. From the multilevel regression, the individual variables: holder status, ages between 35 and 49, and the interaction between gender and age for ages between 40 and 54 were statistically significant at 95% level. We remark that the major differences in waiting times between women and men were observed for individuals between ages from 40 to 54, with women waiting significantly longer.

we all wait the same?", https://doi.org/10.7910/DVN/JYXZYB, Harvard Dataverse.

**Funding:** This research was financed by Complex Engineering Systems Institute (ANID PIA/Apoyo AFB180003).

**Competing interests:** The authors have declared that no competing interests exist.

## Conclusion

Results show the existence of bias in the timeliness of treatment, proving that universal guarantees are not enough to reduce gender inequalities in health care.

## Introduction

It is well known that women and men differ in terms of risks, symptoms and how each group, physically and mentally, experiences medical issues [1–3]. Empirical data show that women have a longer life expectancy than men, but suffer from more illnesses and therefore higher morbidity rates. This gap varies by specific disease and lifecycle stage; see, for example, [4] for a more complete discussion. These differences, although mainly biological, are also affected by socioeconomic and behavioral factors [5]. In one macrolevel study in Denmark, Finland, Norway and Sweden, researchers found that women working outside the home have fewer illnesses and lower hospitalization rates than those who stay at home [6].

Several studies in this field have shown that gender matters and have made considerable contributions to recognizing and understanding gender-based differences in health and healthcare [7–9].

Healthcare coverage, insurance prices, and reimbursements have been studied from a gender-based perspective in different countries. For instance, [10] studied whether there were gender-based differences in medical care in Polish regions using panel data regression. The author concluded that medical reimbursement for some services was higher for men than for women, more women than men reported difficulties in receiving care, and voluntary private insurance was twice as high for men as for women. In [11]'s observational study of clinical referral appointments for tertiary care in a public hospital in India, the authors found gender discrimination in access to healthcare. In [12] the authors study gender-based utilization factors of primary health centers in Pakistan, finding statistical differences in predisposing, enabling and need factors. This discrimination was worse for younger and older patients and for those who lived at increasing distances from the hospital.

There is evidence of explicit and implicit biases regarding gender and career roles among healthcare providers that could affect the treatment of women and men [13]. There is also evidence of patient-physician gender concordance affecting the healthcare outcome, and therefore, bias in regards to women's career roles may have an impact on patients' outcomes [14].

In [15], the authors present a theoretical model that helps frame the sources of potential bias among physicians. The model is based on different assumptions about women and men regarding the level of equity/inequity and sameness/difference between them and concludes that acknowledging differences between men and women is not enough to prevent bias.

There are several studies addressing Chilean healthcare from a gender perspective. The study in [16] presents an overview of the healthcare system in the country, addressing how the coexistence of two separate systems, one with public funds and providers and another with private insurers and providers, has created differences in access for women and men. There have been efforts to foster attention to gender issues in health policy, with limited impact in terms of improving equality of access [17]. Thus, the Chilean healthcare system shows unintended gender inequalities, such as stratification by gender between the public and private systems, with more women of childbearing age enrolled in the public system [18]. Differences between women and men in terms of insurance premiums, coverage, out-of-pocket payments and usage have been comprehensively studied for the private system [19], showing that women pay

more than men for less coverage. Acknowledging these inequalities, the GES plan was created to even out healthcare coverage, access and opportunities for all Chilean residents, regardless of their health plan. In this context, the aim of this paper is to explore the existence of gender bias within the GES Plan. Particularly, we quantify and characterize the differences in waiting time by gender and further explore other observable categories that might be meaningful for understanding these inequalities. We first present a descriptive analysis of the relevant data and then we evaluate potential differences in waiting time between men and women using multilevel hierarchical regressions.

## The Chilean healthcare system

Chile has a mixed health insurance system, where people can choose between a public and a private health plan. The public insurer, in charge of granting protection and health coverage to its beneficiaries and to all those who lack resources, is the National Health Fund (FONASA—Fondo Nacional de Salud), which provides benefits to approximately 75% of the Chilean population. Some 18% of the population is enrolled in a private health plan with one of the several health insurance companies operating in the country, while the rest is either covered under other insurance systems, such as the one provided to the military, or has no coverage at all. The private system offers a variety of plans with different levels of coverage and insurance premiums, while the public system aims to provide universal coverage using the public network of providers.

The public health network is organized into 29 health districts across the country; each administers a group of providers of different complexity within a geographic area [20]. The network of public providers ranges from primary care centers to high-complexity hospitals. In general, patients access the system through lower complexity centers or emergency departments and move to more complex institutions depending on their needs within the assigned health district. For some conditions, there exist some referral centers that not only receive patients from their own health district but also provide coverage at a regional, supra regional and, in some cases, national level [21].

In 2000, Chile introduced an extensive reform of the healthcare system, aimed at achieving a more equitable and fairer system for all citizens, with a law that took effect in August 2004 [22]. This reform, initially known as AUGE (Acceso Universal de Garantías Explícitas) and later renamed the GES plan (Garantías Explícitas en Salud), consists of a set of guarantees regarding access, quality, opportunity, and financial protection for all Chilean citizens diagnosed with any of an established set of diseases (currently 84 conditions). When a patient receives a diagnosis of one of these diseases, she/he is entered into a centrally managed database that tracks all subsequent medical attention. In particular, the opportunity guarantee (OG) included in the reform bill specifies a maximum number of days for receiving appropriate care, ensuring timeliness of the treatments; this opportunity guarantee is the focus of our paper.

If care cannot be provided within the mandated time frame, the insurer has two days to identify an alternative provider who is able to meet the mandate, at no additional cost to the patient. The existence of these guarantees provides a unique opportunity to study, at the patient level, potential gender differences in the delivery of healthcare treatments.

## Materials and methods

### Ethics statement

This work used data from FONASA, which was obtained through a Chilean law –Chile Transparente– that allows the use of deidentified information regarding public programs.

## Data

We use a deidentified database, which includes all cases for the 80 GES diseases and FONASA beneficiaries, with information for the period January 2014 to October 2019. The database consists of 10,876,711 cases, corresponding to transactions generating fund transfers from FONASA to the provider. In the complete database, there are 1,472 healthcare providers, classified into 13 categories according to their degree of complexity and role in the public health network.

For each beneficiary, we have a unique code, date of birth, gender and the insurance holder status (if patient is the primary insurance holder or a dependent). Similarly, for each provider, we have its name, district, region, assigned health district, and category. Finally, for each transaction, we have the beneficiary code, date, tariff, provider of origin, provider of destination, GES disease and specific provision, such as suspicion, confirmation of diagnosis, treatment, and follow-ups.

Given the focus of our study, we exclude diseases that exclusively affect males such as prostate cancer and diseases with a minimum incidence in men, such as breast cancer. We also excluded infant and child diseases from the study.

We construct the waiting time (WT) as the time lapse between two interactions related to a specific OG. It is worth noticing that not all diseases have the same guarantees, both in terms of the maximum time allowed and also the care pathway that is covered. While for some diseases there are OG for diagnostic confirmation, treatment, and follow-up, others include only treatment. Moreover, there are diseases in the GES plan for which the diagnostic confirmations do not generate an additional monetary transfer from FONASA to the provider, and are therefore not recorded in the database, preventing the construction of the WT (for instance the diagnostic confirmation of depression).

Considering all these limitations, we were able to construct WT for fifteen diseases and sixteen GES–OG that are described in Table 1.

## Statistical analysis

We start by performing a descriptive analysis of the database, including the number of cases for each GES–OG, and the percentage of compliance with respect to the opportunity guarantee. We also present the distribution of cases by type of provider, including the percentage of females and primary insurance holders.

Next, we perform a mean test analysis to explore the possibility of differences in WT for female and male patients.

Finally, we use multilevel hierarchical regressions to explain potential differences in WT, considering that our observations fall into clearly identified clusters, such as type of disease, health regional districts and type of providers. Furthermore, multilevel regressions provide a technically robust framework to analyze the correlated nature of the outcome variable [23, 24].

We analyze the influence of aggregate data and individual data on our dependent variable WT using a 4-level regression model. We adopt a modeling strategy that consists of increasing model complexity at each step, using a random intercept model. First, the empty model, without explanatory variables, enabled us to understand the heterogeneity among GES–OG, type of providers and health districts. We test the null hypothesis according to which the variance of the random intercept is zero.

After testing the aggregate levels, we start adding the individual explanatory variables, to account for gender, insurance holder status (primary holder or dependent) and age. Next, we consider GES–OG as the first aggregated level, and finally we check two different model specifications nesting into type of providers and then health districts across the country and the

**Table 1. Diseases and their opportunity guarantees considered in the study.**

| GES–OG | Disease | Description OG | OG [days] |
|---|---|---|---|
| GES17-a | Lymphomas ($\geq$ 15y) | Diagnostic confirmation to staging | 30 |
| GES17-b | Lymphomas ($\geq$ 15y) | Staging to chemotherapy [1] | 10 |
| GES25 | Impulse generation and driving disorders requiring a pacemaker ($\geq$ 15y) | Diagnostic confirmation to treatment | 30 |
| GES26 | Preventive cholecystectomy ($\geq$ 35y and $\leq$ 49y) | Diagnostic confirmation to surgery | 90 |
| GES27 | Gastric cancer | Diagnostic confirmation to treatment | 30 |
| GES31 | Diabetic retinopathy | Diagnostic confirmation to treatment | 60 |
| GES32 | Nontraumatic rheumatogenous retinal detachment | Diagnostic confirmation to treatment | 7 |
| GES37 | Ischemic stroke ($\geq$ 15y) | Diagnostic confirmation to treatment | 1 |
| GES42 | Subarachnoid hemorrhage secondary to ruptured brain aneurysms | Diagnostic confirmation to treatment | 1 |
| GES43 | Primary tumors of the central nervous system | Diagnostic confirmation to treatment | 30 |
| GES49 | Moderate or severe head trauma | Diagnostic confirmation to treatment | 1 |
| GES50 | Severe eye trauma | Diagnostic confirmation to treatment | 3 |
| GES67 | Recurrent remitting multiple sclerosis | Diagnostic confirmation to treatment | 30 |
| GES69 | Hepatitis C | Diagnostic confirmation to pre-treatment | 30 |
| GES70 | Colorectal cancer ($\geq$ 15y) | Diagnostic confirmation to staging | 45 |
| GES73 | Osteosarcoma ($\geq$ 15y) | Medical indication to treatment [2] | 30 |

OG: opportunity guarantee.

GES–OG: specific opportunity guarantee for a given disease.

[1] Staging is the process through examinations and tests to learn the extent of the cancer within the body.

[2] We assume medical indication is equivalent to diagnostic confirmation.

other way around. The contribution to the variance by the stepwise introduction of the different variables (individual and aggregate) in the models was determined by ANOVA test. All tests performed were measured with a statistical significance of 95%.

The specification of the model used is given by Eq 1 with individual explanatory variables (level 1), residing within GES–OG (level 2), nested into type of providers (level 3) and finally nested into health districts (level 4).

$$y_{ijkl} = \quad \beta_0 + \beta_1 x_{1ijkl} + \beta_2 x_{2ijkl} + \sum_{n=1}^{16} \beta_3^n x_{3ijkl}^n +$$

$$\sum_{n=1}^{16} \beta_4^n x_{2ijkl} x_{3ijkl}^n + u_j + u_{jk} + u_{jkl} + \epsilon_{ijkl}$$

(1)

In this model equation $i$ denotes individuals, $j$ denotes GES–OG, $k$ denotes type of provider and $l$ denotes health district. We define the variables of the model as follows:

- $x_{1ijkl}$ = binary variable with 1 if the individual is female, or 0 otherwise, $\forall i, j, k, l$.

- $x_{2ijkl}$ = binary variable with 1 if the individual is the primary insurance holder, or 0 otherwise, $\forall i, j, k, l$.

- $x_{3ijkl}^n$ = binary variable, with 1 if the individual belongs to age group $n$, and 0 otherwise. Age group $n$ corresponds to ages $[5n, 5n + 4]$, $n = 1, \ldots, 16$, with last group for ages $\geq$ 80, $\forall i, j, k, l$, $\forall n, i, j, k, l$. The use of discrete age variables allows to capture non-linear effects.

- $\beta_0$ = intercept.

- $\beta_1$ = coefficient for the individual variable $x_{1ijkl}$.

- $\beta_2$ = coefficient for the individual variable $x_{2ijkl}$.

- $\beta_3^n$ = coefficient for the individual variable $x_{3nijkl}$.

- $\beta_4^n$ = coefficient for the interaction between variables $x_{2ijkl}$ and $x_{3nijkl}$.

- $\epsilon_{ijkl}$ = residual effect at individual level within our aggregate levels (GES–OG, type of provider and health district).

- $u_j$ = specific effect (deviation from $\beta_0$) of GES–OG.

- $u_{jk}$ = specific effect of type of provider within GES–OG.

- $u_{jkl}$ = specific effect of health district within type of provider within GES–OG.

All statistical analyses were performed using programs: R version 3.6.3 and R-Studio version 1.2.5033.

## Results

### Database descriptive analysis

Table 2 shows a summary of the main characteristics of the dataset. There are 730,783 cases of which 33.5% of them have information to compute the WT, corresponding to 244,812 cases.

We observe that although the GES plan guarantees a maximum WT for each disease, compliance ranges from 98.4% (GES49—moderate or severe head trauma) to only 38.4% (GES27 —gastric cancer), with an average compliance of 77.4% for the dataset. Moreover, the excess waiting time for most diseases is more than double the OG, when the latter is not satisfied.

Table 3 reports the distribution of cases among the type of provider. We first observe that 5 categories, high-, medium-, and low-complexity providers, health referece centers, and health centers, cover over 99.9% of the cases in the dataset. The type of provider considers the resolution capacity, the infrastructure and the degree of specialization of the workforce. High complexity providers are larger hospitals (more than 300 beds) with 20% of its bed capacity assigned to critical care. They provide access to all (or most) subspecialties and diagnostic and treatment equipments. Medium complexity providers are hospitals with 31 to 300 beds, with a small proportion of beds for critical care, if any. They have lower level of infrastructure and equipments (no radiotherapy or chemotherapy, for instance) and they do not have all the subspecialties staffed. Low complexity providers are small hospitals with up to 30 beds, delivering only basic care. Health centers and health reference centers are medium-complexity institutions that provide only outpatient health services. However, they are responsible for referring patients within their health district who need care exceeding the center's level of complexity. In such cases, the institution pays for treatment and receives the FONASA reimbursement.

The data set contains 48.7% of females and most of the cases are associated to primary insurance holders, with an average of 94.3%. The distribution of cases by age group and health district can be found in Figs 1 and 2.

### Mean test analysis

Table 4 shows the results of the Welch two-sample t-test for each GES–OG. We notice that the Welch test is suitable for our case, since our sample has unequal variances and sample sizes. The null hypothesis ($H_0$) states that there is no difference between the average waiting time for males and females. The alternative hypothesis ($H_1$) is that the average waiting time for men is smaller than that for women, except for GES25 (impulse generation and driving disorders

**Table 2. Descriptive analysis of dataset.**

| GES–OG | Number of cases in dataset | Number of cases under study (%) | | $\overline{WT}$(SD) [days] | | Compliance [%] | $\overline{WT}$ ($\overline{WT}$>OG) [days] |
|---|---|---|---|---|---|---|---|
| GES17-a | 10,403 | 5,711 | (54.9) | 12.8 | (25.7) | 86.8 | 57.6 |
| GES17-b | 8,016 | 5,565 | (69.4) | 25.9 | (98.5) | 62.3 | 63.6 |
| GES25 | 34,513 | 25,452 | (73.7) | 12.3 | (33.1) | 90.4 | 62.4 |
| GES26 | 106,590 | 25,652 | (24.1) | 71.9 | (141.9) | 77.4 | 206.9 |
| GES27 | 63,634 | 5,334 | (8.4) | 69.8 | (116.1) | 38.4 | 103.2 |
| GES31 | 129,624 | 35,749 | (27.6) | 126.9 | (219.9) | 57.6 | 263.8 |
| GES32 | 13,601 | 6,428 | (47.3) | 10.1 | (38.7) | 71.9 | 25.5 |
| GES37 | 156,667 | 95,289 | (60.8) | 1.3 | (25.9) | 78.1 | 13.8 |
| GES42 | 4,704 | 2,362 | (50.2) | 1.7 | (5.6) | 76.3 | 6.2 |
| GES43 | 7,564 | 2,807 | (37.1) | 68.0 | (162.9) | 64.1 | 171.5 |
| GES49 | 89,071 | 19,953 | (22.4) | 0.3 | (10.8) | 98.4 | 15.9 |
| GES50 | 77,053 | 6,378 | (8.3) | 1.3 | (13.3) | 93.7 | 13.2 |
| GES67 | 962 | 490 | (50.9) | 51.7 | (101.7) | 58.1 | 106.5 |
| GES69 | 1,656 | 725 | (43.8) | 23.7 | (91.6) | 85.9 | 153.0 |
| GES70 | 26,539 | 6,827 | (25.7) | 16.5 | (27.5) | 92.3 | 71.3 |
| GES73 | 186 | 90 | (48.4) | 47.3 | (93.9) | 50.0 | 78.1 |
| **Total** | **730,783** | **244,812** | **(33.5)** | | | **77.4** | |

GES–OG: specific opportunity guarantee for a given disease.

$\overline{WT}$: average waiting time, SD: standard deviation of waiting time.

Compliance: fraction of cases where the opportunity guarantee was satisfied.

$\overline{WT}$ | ($\overline{WT}$ > OG): waiting time for those patients where the opportunity guarantee was not satisfied.

requiring a pacemaker), GES37 (ischemic stroke), GES49 (moderate or severe head trauma), GES67 (recurrent remitting multiple sclerosis), GES70 (colorectal cancer), and GES73 (osteo-sarcoma) where the opposite relationship is studied.

For four GES–OG, we reject the null hypothesis of equal means, accepting the alternative hypothesis: the average WT for men is significantly smaller than the WT for women. This result holds for GES17-a, GES17-b (lymphomas), GES26 (preventive cholecystectomy), GES31 (diabetic retinopathy) and GES32 (nontraumatic rheumatogenous retinal detachment). For another two GES–OG, we reject the null hypothesis of equal means, accepting the alternative hypothesis that the average WT for men is significantly larger than that for women: GES37

**Table 3. Distribution of cases by type of provider, percentage of female patients and primary insurance holders.**

| Type of provider | Number of cases | Percentage over total cases [%] | Female [%] | Primary insurance holder [%] |
|---|---|---|---|---|
| High-Complexity | 222,109 | 90.7 | 48.2 | 94.3 |
| Medium-Complexity | 5,895 | 2.4 | 60.8 | 94.5 |
| Low-Complexity | 2,753 | 1.1 | 54.2 | 94.5 |
| Health Center | 10,710 | 4.4 | 51.0 | 94.0 |
| Health Reference Center | 3,257 | 1.3 | 50.5 | 92.8 |
| Other | 88 | 0.1 | 81.8 | 90.9 |
| **Total** | **244,812** | **100.0** | **48.7** | **94.3** |

The type of provider considers the resolution capacity, the infrastructure and the degree of specialization of the workforce.

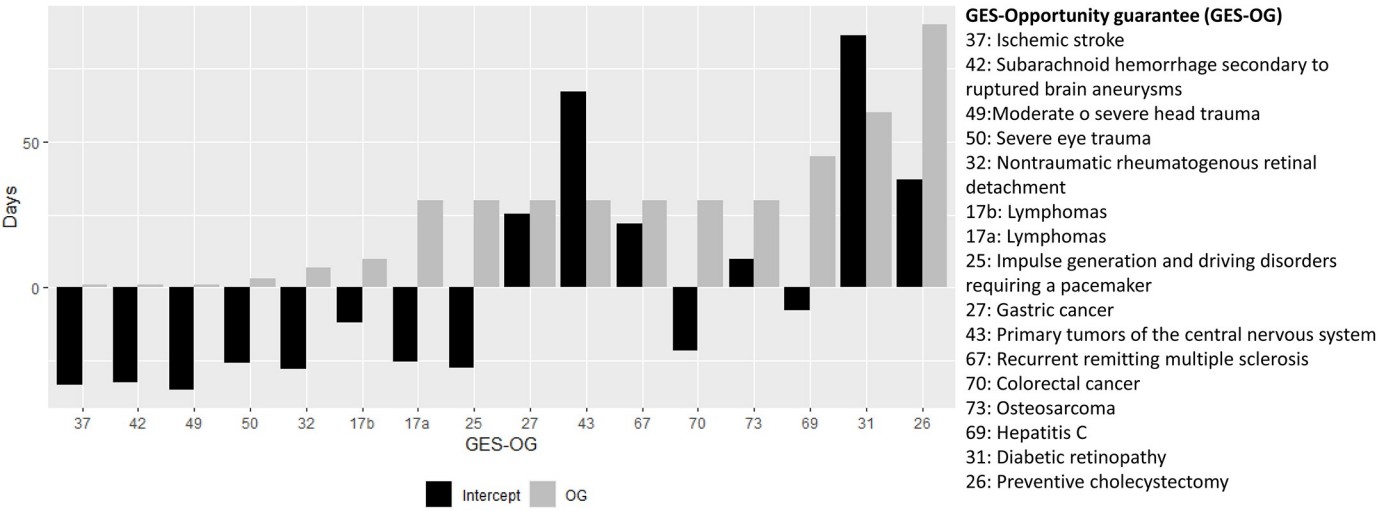

**Fig 1. Multilevel hierarchical regression intercepts at opportunity guarantee level with their corresponding opportunity guarantee.**

(ischemic stroke) and GES70 (colorectal cancer). For the rest of the GES–OG, we cannot reject the null hypothesis of equal WT means.

## Multilevel analysis

We present the results of our final 4–level model, with binary variables for female, primary holder status, age group, and the interaction between female and age group (level 1), residing within GES–OG (level 2), nested into type of provider (level 3) and finally nested into health districts across the country (level 4).

Table 5 shows the results of our 4–level regression. The variable primary holder is statistically significant with positive slope in WT. The variables for ages between 35 to 49 years old are also significant with negative impact in WT. The interactions of gender and ages between 40 to 54 years old are significant with positive impact in the WT.

In Fig 3, we show the intercept at the GES–OG level and its corresponding OG, observing a positive correlation between these two variables, that is stronger for GES–OGs with shorter OG (less than 10 days). Notice that GES–OGs are sorted from shorter to larger OG.

Table 6 shows the intercept for each provider type within the GES–OG level. High complexity providers show, on average, a negative impact on WT, especially for GES–OGs with shorter OG. There are GES–OGs with a large range for intercepts, showing great heterogeneity among different types of providers. See, for example, GES31 (diabetic retinopathy), GES43 (primary tumors of the central nervous system), and GES67 (recurring remitting multiple sclerosis).

There is an increased variability in the intercepts at the type of provider level for GES–OGs with larger OGs. Intercepts for OGs lower than or equal to 10 days (greater than 10 days) have a standard deviation of 4.4 (13.8).

Table 7 shows the intercept for each GES–OG and health district, where the effect of type of provider is included using their weighted averages (the weights are the number of cases for each type of provider). Health districts are sorted from highest to lowest total number of cases in the dataset. Once again, we observe a high variability for GES–OGs with large OGs.

Finally, in Fig 4 we present an example of the prediction of our model for GES–OG 37, in health district 16, for high, medium and low complexity providers. We show the predicted WT

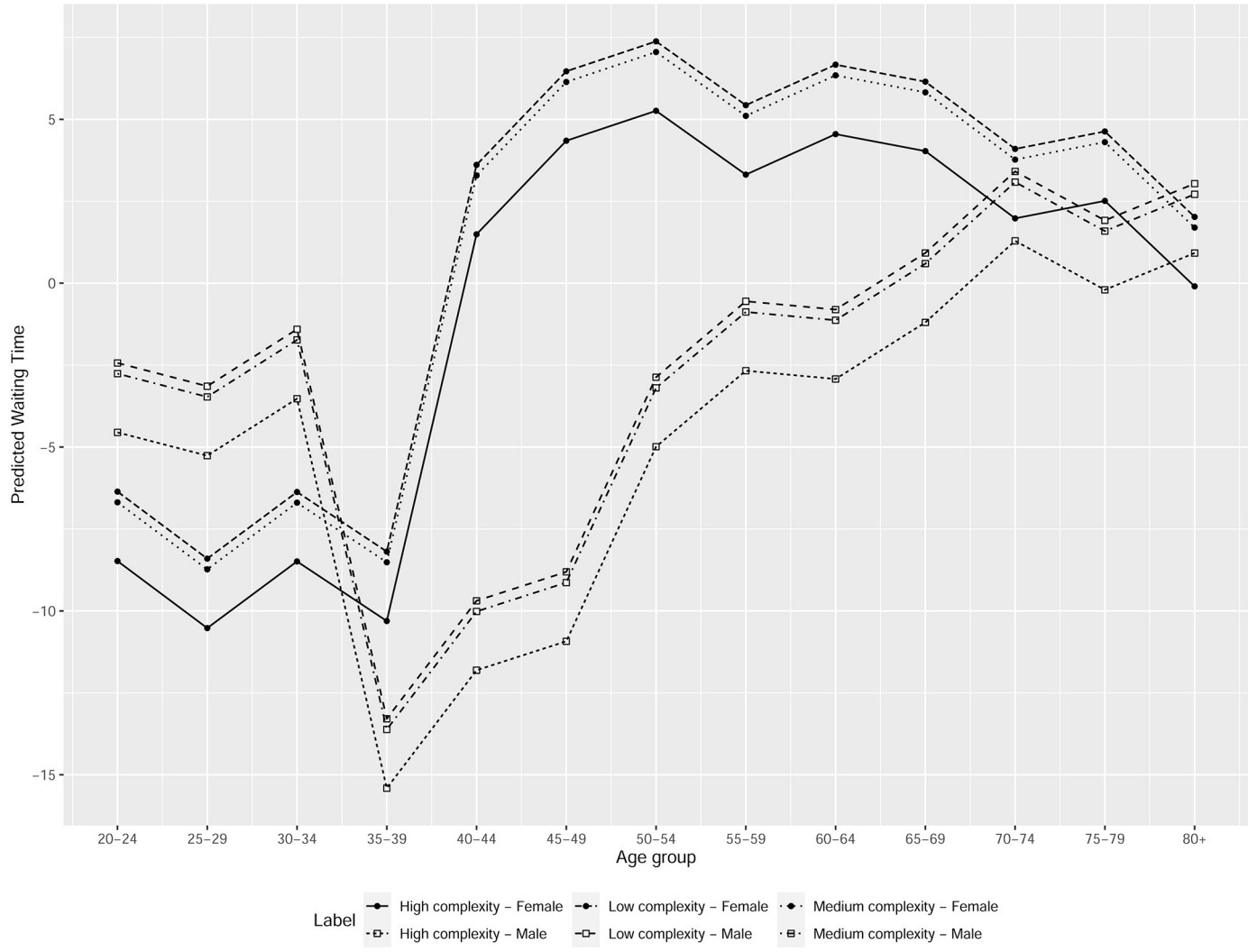

**Fig 2. Example of predicted waiting time differences between female and male.**

for women and men as a function of age. We observe that the predicted WT for women is larger than that for men for ages between 35 and 65 years old, for the 3 provider types.

## Discussion

This study intended to characterize WT for certain diseases, using the FONASA database and assessed the existence of gender bias in this WT. We explored how WT is affected by characteristics such as age, insurer holder status, health district, and type of provider assigned. Most of the literature on gender bias in healthcare studies the differences in access, use and levels of insurance of the population. To the best of our knowledge, this is the first study in Chile, that analyzes the timeliness of treatment after a diagnosis has been made.

We found significant differences in WT for treatment between men and women for 7 out of the 16 GES–OGs analyzed. In two of them, GES 37 (ischemic stroke) and GES70 (colorectal cancer), the WT was shorter for women than men. In GES37, women had a WT 0.3 days shorter than men, and even though it might be a small difference, due to the characteristics of the disease, it may have an impact on the health outcome, that needs further study. We also

**Table 4. Welch two-sample t-test for average waiting time between male and female patients.**

| GES–OG | N° male cases | N° female cases | $\overline{WT}_m$ (SD) | | $\overline{WT}_f$ (SD) | | p-value | |
|---|---|---|---|---|---|---|---|---|
| GES17-a | 2,957 | 2,754 | 12.2 | (25.4) | 13.5 | (26.0) | 0.0238 | (**) |
| GES17-b | 2,923 | 2,642 | 23.6 | (85.5) | 28.6 | (111.8) | 0.0294 | (**) |
| GES25[(1)] | 13,759 | 11,693 | 12.4 | (32.4) | 12.1 | (34.0) | 0.2491 | |
| GES26 | 5,599 | 20,053 | 47.3 | (112.1) | 79.0 | (148.7) | $\leq 0.001$ | (***) |
| GES27 | 3,618 | 1,716 | 68.9 | (107.5) | 71.8 | (132.5) | 0.2193 | |
| GES31 | 17,068 | 18,681 | 119.9 | (208.4) | 133.4 | (229.8) | $\leq 0.001$ | (***) |
| GES32 | 3,581 | 2,847 | 9.5 | (41.2) | 10.8 | (35.3) | 0.0913 | (*) |
| GES37[(1)] | 50,663 | 44,626 | 1.4 | (28.4) | 1.1 | (22.7) | 0.0551 | (*) |
| GES42 | 652 | 1,710 | 1.6 | (4.5) | 1.8 | (6.0) | 0.2268 | |
| GES43 | 1,014 | 1,793 | 66.8 | (157.9) | 69.0 | (166.2) | 0.3666 | |
| GES49[(1)] | 14,646 | 5,307 | 0.4 | (12.3) | 0.3 | (5.0) | 0.1953 | |
| GES50 | 5,032 | 1,346 | 1.3 | (14.4) | 1.4 | (7.8) | 0.3162 | |
| GES67[(1)] | 151 | 339 | 56.8 | (121.2) | 49.6 | (92.3) | 0.2559 | |
| GES69 | 409 | 316 | 22.1 | (85.0) | 26.2 | (100.8) | 0.2784 | |
| GES70[(1)] | 3,475 | 3,352 | 17.1 | (29.0) | 15.9 | (25.8) | 0.0375 | (**) |
| GES73[(1)] | 55 | 35 | 53.2 | (119.8) | 38.2 | (30.7) | 0.1897 | |

GES–OG: specific opportunity guarantee for a given disease.

$\overline{WT}_{m/f}$: average waiting time for males/females.

$H_0 : \overline{WT}_m - \overline{WT}_f = 0; H_1 : \overline{WT}_m - \overline{WT}_f < 0;$

[(1)] $H_1 : \overline{WT}_m - \overline{WT}_f > 0$

(*), (**), and (***): significant at 90%, 95%, and 99%, respectively.

observe that GES37 accounts for 50,663 cases, and therefore, due to the large amount of data, small differences will almost certainly be statistically significant. On the other hand, for GES70, colorectal cancer, WT for women is 1.2 days shorter than that for men. However, this difference will most likely not have an impact on health outcomes, considering the OG is 45 days.

For GES26 (preventive cholecystectomy) and GES31 (diabetic retinopathy), the t-test p-values are smaller than 0.001, and therefore, there is strong evidence that women wait longer than men in these two pathologies. We notice that for GES26, women account for almost 80% of the cases, with an average WT 1.67 times longer than that for men (47.3 vs 79.0 days). For GES31, women account for almost half of the cases and wait, on average, 2 weeks longer to receive the appropriate treatment. Diabetic retinopathy produces slow progressive eye damage, but it might also cause sudden complications such as vitreous hemorrhage, retinal detachment and blindness. Thus, it would be interesting to study if outcomes for women are worse than those for men, when waiting longer for treatment.

We notice that the opportunity guarantees analyzed in this study vary from one to ninety days. There are five GES–OGs which mandate treatment within less than a week (GES32: non-traumatic rheumatogenous retinal detachment, GES37 (ischemic stroke in people 15 years and older), GES42 (subarachnoid hemorrhage secondary to ruptured brain aneurysms), GES49 (moderate or severe head trauma), and GES50 (severe eye trauma). Therefore, there is less room for discretionary decisions affecting WT given the urgency of these treatments.

Although more than 90% of cases in our database were treated in high complexity hospitals, we observe that the type of provider in our multilevel analysis increased the fit of our model. We found larger variability in the intercept for the type of provider for GES–OG with OGs

**Table 5. Statistical results for the 4-level model.**

**Random effects**:

| Groups | Std.Dev. | Obs. |
|---|---|---|
| Health district nested in type of establishment nested in GES–OG | 28.38 | 523 |
| Type of provider nested in GES–OG | 21.43 | 61 |
| GES–OG | 40.28 | 16 |
| Residual | 100.06 | |

**Fixed effects**:

| | Estimate | SE | p-value | |
|---|---|---|---|---|
| (Intercept) | 38.0700 | 11.3100 | 0.0036 | (***) |
| Female | 0.2825 | 4.1970 | 0.9463 | |
| Primary holder | 2.0410 | 0.9543 | 0.0324 | (**) |
| age [05-09] | -0.7692 | 4.7020 | 0.8700 | |
| age [10-14] | -0.8775 | 4.8930 | 0.8577 | |
| age [15-19] | -1.2630 | 3.8900 | 0.7455 | |
| age [20-24] | -1.7010 | 3.6740 | 0.6433 | |
| age [25-29] | -2.4110 | 3.6260 | 0.5062 | |
| age [30-34] | -0.6724 | 3.5850 | 0.8512 | |
| age [35-39] | -12.5600 | 3.3280 | 0.0002 | (***) |
| age [40-44] | -8.9580 | 3.2310 | 0.0056 | (***) |
| age [45-49] | -8.0750 | 3.1440 | 0.0102 | (**) |
| age [50-54] | -2.1370 | 3.1170 | 0.4931 | |
| age [55-59] | 0.1811 | 3.0770 | 0.9531 | |
| age [60-64] | -0.0719 | 3.0630 | 0.9813 | |
| age [65-69] | 1.6560 | 3.0540 | 0.5877 | |
| age [70-74] | 4.1430 | 3.0550 | 0.1750 | |
| age [75-79] | 2.6490 | 3.0770 | 0.3894 | |
| age [80,+) | 3.7710 | 3.0480 | 0.2160 | |
| Female×age [05-09] | -0.3681 | 7.5000 | 0.9609 | |
| Female×age [10-14] | 0.8260 | 8.6390 | 0.9238 | |
| Female×age [15-19] | -1.6130 | 6.4540 | 0.8027 | |
| Female×age [20-24] | -4.2080 | 5.9060 | 0.4761 | |
| Female×age [25-29] | -5.5440 | 5.6690 | 0.3281 | |
| Female×age [30-34] | -5.2480 | 5.4500 | 0.3356 | |
| Female×age [35-39] | 4.8190 | 4.6300 | 0.2980 | |
| Female×age [40-44] | 13.0200 | 4.5550 | 0.0043 | (***) |
| Female×age [45-49] | 14.9900 | 4.4680 | 0.0008 | (***) |
| Female×age [50-54] | 9.9660 | 4.4880 | 0.0264 | (**) |
| Female×age [55-59] | 5.7000 | 4.4240 | 0.1976 | |
| Female×age [60-64] | 7.1890 | 4.3940 | 0.1018 | |
| Female×age [65-69] | 4.9440 | 4.3740 | 0.2584 | |
| Female×age [70-74] | 0.4037 | 4.3680 | 0.9264 | |
| Female×age [75-79] | 2.4320 | 4.3850 | 0.5793 | |
| Female×age [80,+) | -1.2990 | 4.3170 | 0.7635 | |

GES–OG: specific opportunity guarantee.

Level 1: binary variables for female, primary holder status, age group, and interaction between female and age group, level 2: GES–OG, level 3: type of provider, level 4: health districts across the country.

(*), (**), and (***): significant at 90%, 95%, and 99%, respectively.

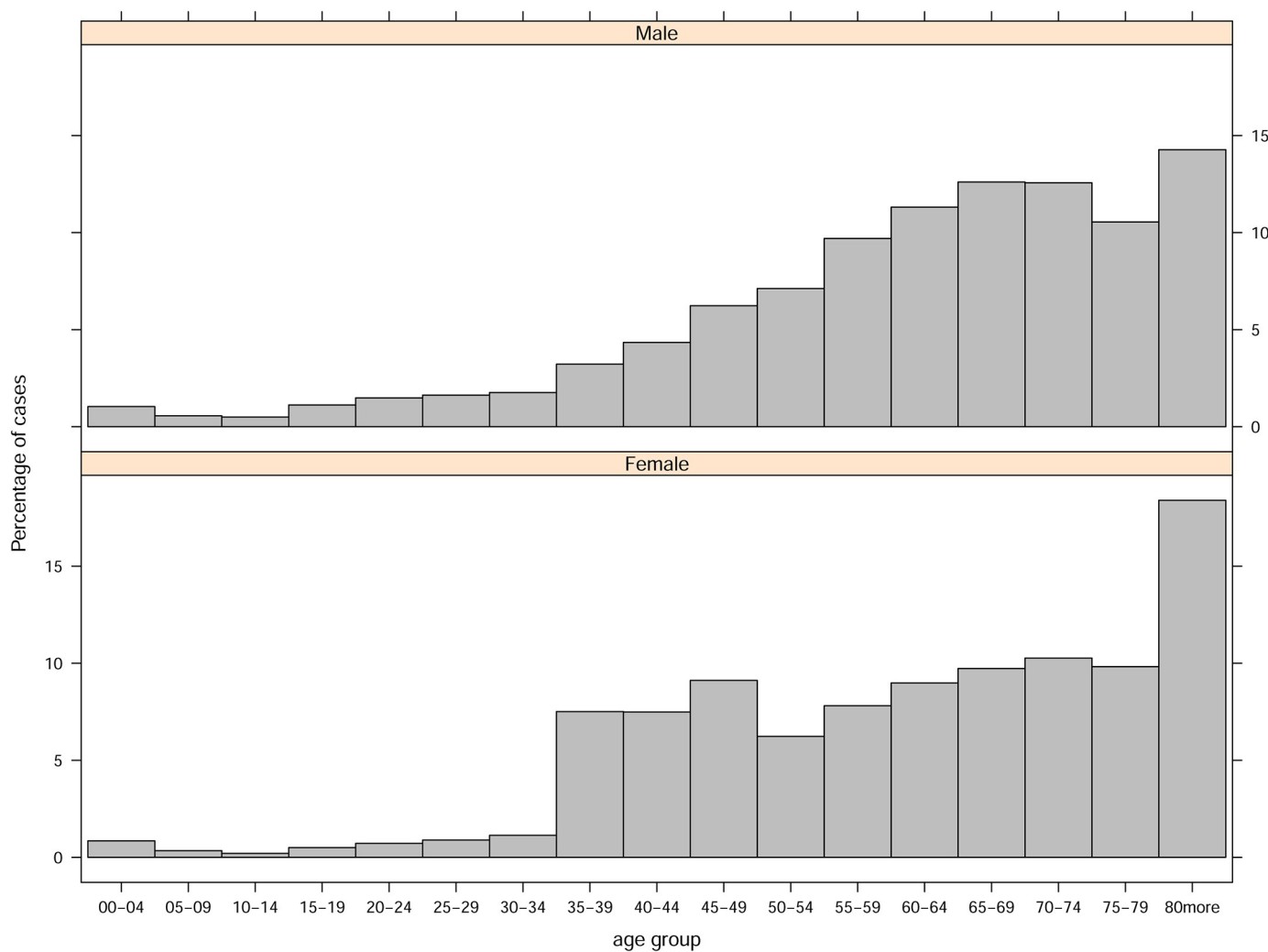

**Fig 3. Distribution of cases by age group and gender.**

**Table 6. Intercept for each provider type within disease opportunity guarantee level.**

|  | GES–OG | | | | | | | | | | | | | | | |
|---|---|---|---|---|---|---|---|---|---|---|---|---|---|---|---|---|
|  | 37 | 42 | 49 | 50 | 32 | 17b | 17a | 25 | 27 | 43 | 67 | 70 | 73 | 69 | 31 | 26 |
| **HC** | -7.8 | -6.9 | -4.0 | -5.3 | -4.1 | -3.7 | -3.0 | -4.3 | 4.1 | -12.3 | -13.3 | -5.3 | 2.4 | 3.4 | 8.0 | -3.4 |
| **MC** | -7.1 |  | -1.3 | -0.5 |  |  |  | -2.1 | -0.1 | 52.4 | 18.7 | -2.5 |  | -1.1 | 2.5 | 15.9 |
| **LC** | -4.9 |  | -1.2 |  | -0.3 |  |  |  |  | -0.7 |  |  |  |  | -12.0 | 21.0 |
| **HeC** | -1.4 | -1.0 | -1.1 |  | -0.9 | 0.8 | -3.2 | 3.2 | 4.6 |  | -0.1 | 1.9 |  | -4.2 | -5.3 | 6.6 |
| **RefC** | 13.0 |  | -0.6 |  | -1.5 |  |  | -2.9 | -2.5 |  |  | -1.0 |  |  | 28.0 | 10.1 |

GES–OG: specific opportunity guarantee for a given disease (sorted from shorter to larger opportunity guarantee).

HC: high complexity, MC: medium complexity, LC: low complexity, HeC: health center, RefC: health reference center.

The intercept for type of provider within disease opportunity guarantee level shows great heterogeneity among different types of providers. Negative/positive intercept values imply a decrease/increase in waiting times by that amount for that disease and that type of provider from the overall average (fixed effects intercept).

**Table 7. Weighted average intercept for the provider within each health district and specific opportunity guarantee.**

| | GES–OG | | | | | | | | | | | | | | | |
|---|---|---|---|---|---|---|---|---|---|---|---|---|---|---|---|---|
| | 37 | 42 | 49 | 50 | 32 | 17b | 17a | 25 | 27 | 43 | 67 | 70 | 73 | 69 | 31 | 26 |
| HA-14 | -0.9 | -1.7 | 0.1 | -0.9 | 2.2 | 1.7 | 29.8 | -0.1 | -21.3 | -32.4 | 4.7 | -6.5 | | 3.2 | 15.4 | -35.7 |
| HA-12 | -1.3 | -3.0 | -1.1 | -6.2 | -4.5 | 0.2 | 1.2 | 3.0 | -0.4 | -44.1 | 17.4 | -1.1 | | -25.5 | 65.9 | -16.7 |
| HA-16 | -2.1 | -0.7 | -0.1 | -4.3 | -2.6 | 0.6 | 3.6 | -1.4 | -12.0 | -55.5 | -14.7 | -3.1 | | -2.9 | -29.0 | 2.3 |
| HA-21 | -0.8 | -1.4 | 0.0 | -2.2 | -0.9 | -15.2 | -8.7 | -1.5 | 1.6 | 1.9 | 2.6 | 11.4 | | 2.4 | 62.4 | -5.5 |
| HA-10 | -1.7 | -0.6 | -0.9 | | 7.2 | 9.1 | -2.5 | 3.6 | -4.4 | -60.1 | 1.8 | -0.4 | 20.6 | -31.9 | -16.1 | -28.8 |
| HA-7 | -0.7 | | -0.3 | -5.1 | 2.0 | 0.4 | -7.4 | 3.3 | -11.4 | -61.3 | -7.9 | -1.1 | | | -43.5 | -16.7 |
| HA-15 | -0.9 | -0.3 | -0.9 | 0.2 | -2.6 | 42.0 | 2.4 | -4.0 | -13.9 | 16.8 | 11.6 | 5.7 | | | -20.3 | 39.0 |
| HA-13 | -4.0 | -1.6 | -0.7 | -4.3 | -0.5 | 1.0 | -7.6 | 8.5 | -4.2 | 5.5 | 22.1 | 2.0 | | -12.6 | -19.8 | 25.0 |
| HA-17 | -1.6 | -0.3 | -0.2 | -6.9 | 0.5 | -4.3 | 12.3 | 0.6 | 14.6 | 1.8 | 20.0 | 5.1 | | -4.2 | -32.3 | 21.0 |
| HA-18 | -1.0 | -1.2 | -0.1 | -3.6 | -0.7 | -5.2 | 4.2 | -2.7 | 18.6 | -21.5 | -18.1 | 14.7 | | -29.5 | 50.9 | 9.9 |
| HA-6 | 1.5 | -2.1 | -0.3 | -5.1 | 9.2 | -14.2 | -5.2 | 5.3 | 4.5 | -7.7 | -4.2 | -2.6 | | 42.0 | 20.1 | 14.4 |
| HA-11 | -0.8 | -0.4 | -1.0 | -0.5 | -2.1 | -1.4 | -5.8 | 3.5 | 13.1 | 96.2 | -4.4 | -5.3 | | 16.3 | -80.9 | -30.4 |
| HA-5 | -1.7 | -2.4 | 0.1 | -5.5 | 0.6 | 9.7 | 7.9 | -5.7 | -5.6 | 29.3 | -4.2 | -2.1 | | 40.0 | -24.2 | 38.5 |
| HA-22 | -1.9 | -1.1 | -0.4 | -0.8 | -1.6 | -9.1 | -12.0 | 7.2 | 16.3 | 47.9 | -27.5 | 2.8 | | -11.7 | 43.5 | 37.7 |
| HA-24 | -0.4 | -3.1 | -0.2 | -6.0 | 0.0 | | 2.6 | -0.4 | -10.7 | -49.4 | -10.6 | 0.3 | | 36.1 | 67.0 | -14.4 |
| HA-19 | -1.1 | | -0.4 | -1.4 | -2.0 | -13.0 | -2.2 | -10.9 | 12.8 | -37.0 | 11.7 | 12.2 | | -8.5 | 23.8 | 4.7 |
| HA-9 | 1.4 | | -0.3 | | 0.6 | 7.9 | 11.2 | -0.3 | 3.7 | -60.7 | -10.1 | 5.6 | -15.8 | 2.3 | 12.4 | -23.6 |
| HA-3 | -0.2 | 2.0 | 0.0 | -4.7 | -3.4 | 31.9 | 0.2 | -3.4 | -3.5 | 2.9 | -1.0 | -1.8 | | -20.5 | -75.7 | -18.8 |
| HA-20 | -2.0 | 1.9 | -0.6 | 71.6 | -0.9 | -12.8 | -11.7 | -6.3 | -18.3 | -21.2 | -11.2 | -5.8 | | 2.1 | 21.4 | 28.5 |
| HA-23 | -1.4 | | -0.8 | -5.5 | -0.7 | -9.0 | -2.8 | -5.5 | 9.7 | -25.0 | 6.0 | -6.1 | | -5.9 | -71.9 | -10.6 |
| HA-8 | 0.1 | | -0.5 | | -0.7 | -2.2 | -10.8 | 6.9 | -31.2 | 124.1 | 23.2 | -1.0 | | -8.3 | -4.4 | 19.7 |
| HA-4 | -1.1 | 1.2 | 0.0 | | | | | -2.2 | 0.8 | 61.8 | -10.2 | -5.6 | | -4.5 | -59.7 | -35.8 |
| HA-1 | -1.4 | | -0.7 | -3.5 | -3.9 | -16.1 | -11.0 | -5.5 | -20.4 | | | -4.6 | | -6.0 | 66.8 | -47.9 |
| HA-29 | 1.2 | | -1.2 | -4.7 | | | | | -2.8 | | 0.4 | -2.3 | | -7.5 | 14.3 | 4.0 |
| HA-2 | 0.2 | | 0.7 | -1.4 | | | | 3.9 | 70.6 | -44.4 | | -9.4 | | 36.1 | -28.0 | -32.6 |
| HA-26 | -0.3 | 1.0 | -0.2 | -5.1 | -4.6 | -6.5 | 1.4 | -7.9 | 2.0 | -14.2 | -11.0 | 6.3 | | | -33.5 | -13.3 |
| HA-28 | 5.9 | | | | -0.5 | | | | | -1.4 | | | | | -23.8 | 21.0 |
| HA-33 | -0.5 | | 1.8 | | | | | | -25.0 | 20.2 | -0.3 | -5.8 | | | 63.7 | 9.0 |
| HA-25 | -1.1 | | 0.0 | -4.5 | | -6.4 | -3.0 | 0.0 | 29.7 | | -6.9 | -1.9 | | | 6.4 | 7.8 |

HA: health district.

GES–OG: specific opportunity guarantee for a given disease (sorted from shorter to larger opportunity guarantee).

Intercept for each disease opportunity guarantee and health district, where the effect of type of provider is included using their weighted averages. Health districts are sorted from highest to lowest total number of cases in the dataset.

longer than 10 days, that may be a consequence of more room for discretionality due to the length of the OG. We also found large heterogeneity across health districts in their impact on WT for each GES–OG.

At the individual level, primary holder status and age groups between 35 and 49, were significant when explaining WT. These two variables are associated with active workers, and we might think they would affect waiting time in the same direction. However, this is not the case: primary insurance holders have a positive slope in WT, and ages between 35 and 49 have a negative one. It is worth noticing that the coefficient for ages is at least 4 times larger than the one for the primary holder status variable.

For women, the coefficient was not statistically significant. However, in the interaction between the gender and age variables we found an interesting and revealing effect: for female

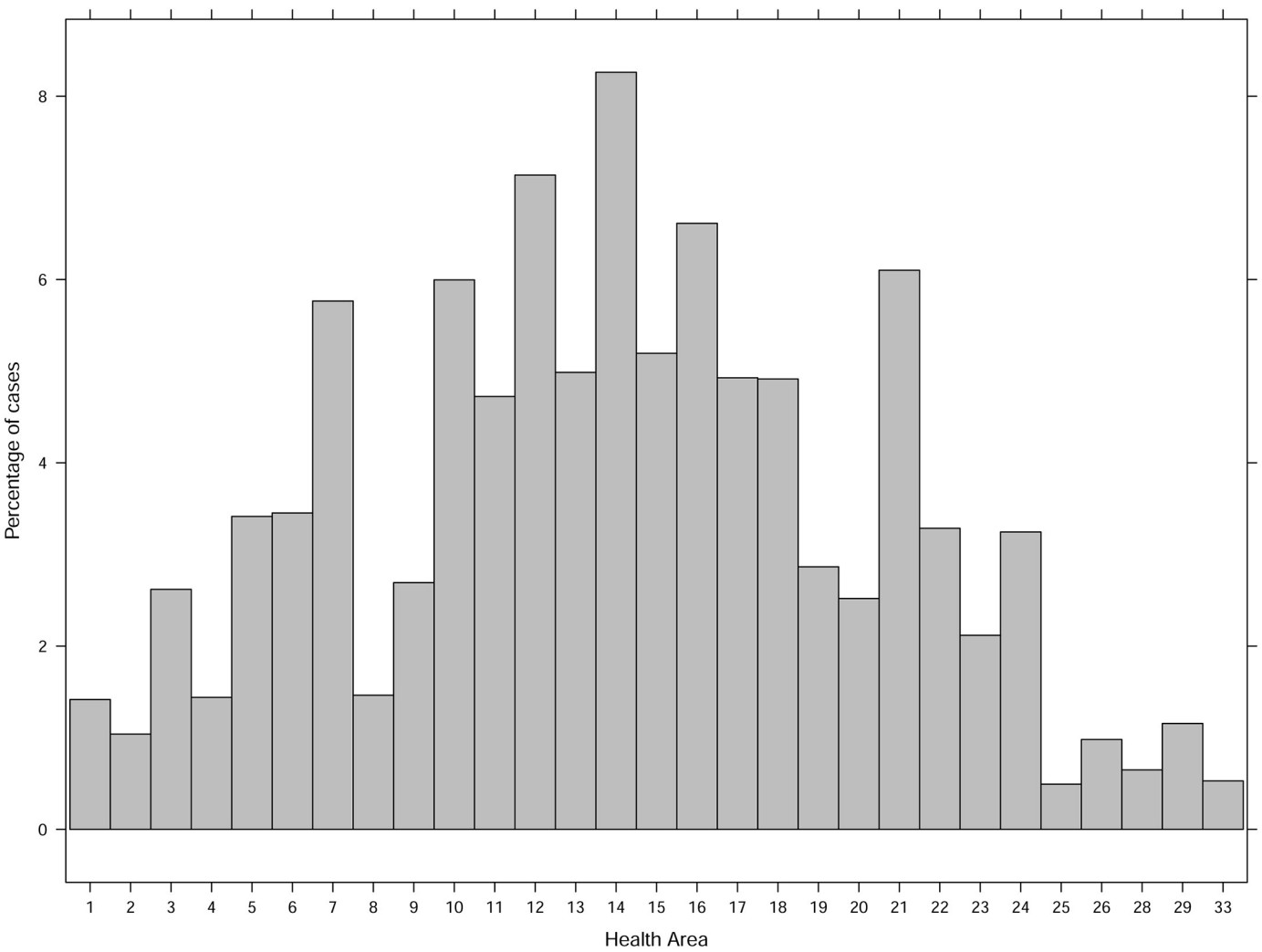

**Fig 4. Distribution of cases by health district.**

patients between 40 and 54 years old the impact on WT is significant and positive, more than compensating the negative effect on WT that we found for working ages. We should further explore the reasons behind this bias.

Although not part of the focus of our study, we found a relatively low rate of OG compliance, with an average of 77.4% for the 16 GES–OGs under study. We also noticed a high degree of WT variability within each GES–OG, with coefficients of variation (std/average WT) that go from 1.7 to 36. It would be important, in further research, to explore the sources of these variabilities that might reflect heterogeneity and discretionalities, both from patients and system administrators.

The main limitation of this study is that we did not explicitly consider the congestion of the healthcare system. Including the latter would be useful to understand compliance rates and their possible impact in WT. Further research is needed to understand if under the pressure of congestion, gender bias is exacerbated.

## Conclusion

The existence of a database with timestamps for each step of health service delivery at an individual level allowed us to perform a detailed analysis regarding potential gender bias regarding waiting times for treatment in Chile. We note that this type of data is not only important at a management level but also crucial to evaluate the impact of public health policies.

Our analysis shows that the existence of explicit opportunity guarantees for GES–OG does not prevent bias when considering the timeliness of treatment between women and men. This bias is impacted by the difference among providers and health districts, along with other observable patients characteristics such as age and insurance holder status. We believe, with the limited evidence at hand, that differences in waiting times are most likely a product of a complex combination of several factors, where the role of women in society might be a fundamental component. Understanding these factors is part of our ongoing research. Once the reasons behind these biases are known, more specific, differentiated, gender-oriented policies should be implemented. In the meantime, positive actions that facilitate timely treatment for women should be considered, especially for those between 40 and 54 years old.

## Acknowledgments

We thank Francisco León, Director of Institutional Planning, Germán Vera, Head of Research and Statistics, and Viviana Ulloa, statistical analyst at the Chilean Fondo Nacional de Salud (FONASA), for providing the databases used in this study and for their patience in answering all our questions. Without their help, this work would not have been possible.

## Author Contributions

**Conceptualization:** Susana Mondschein, Natalia Yankovic.

**Data curation:** Maria Quinteros.

**Formal analysis:** Susana Mondschein, Maria Quinteros, Natalia Yankovic.

**Investigation:** Susana Mondschein.

**Methodology:** Susana Mondschein, Maria Quinteros, Natalia Yankovic.

**Project administration:** Susana Mondschein.

**Resources:** Susana Mondschein.

**Supervision:** Susana Mondschein, Natalia Yankovic.

**Validation:** Susana Mondschein, Natalia Yankovic.

**Writing – original draft:** Susana Mondschein, Natalia Yankovic.

**Writing – review & editing:** Susana Mondschein, Natalia Yankovic.

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
