## [Decision Letter · Decision Letter 0]

24 Apr 2020

PONE-D-20-03609

Bias in the Chilean public health system: Do we all wait the same?

PLOS ONE

Dear Dr. mondschein,

Thank you for submitting your manuscript to PLOS ONE. After careful consideration, we feel that it has merit but does not fully meet PLOS ONE’s publication criteria as it currently stands. Therefore, we invite you to submit a revised version of the manuscript that addresses the points raised during the review process.

Both reviewers raised a number of considerations that should be addressed, particularly the application of a more scientific structure to the manuscript, addressing more relevant and global literature, and providing policy implications of the findings.

We would appreciate receiving your revised manuscript by Jun 08 2020 11:59PM. To enhance the reproducibility of your results, we recommend that if applicable you deposit your laboratory protocols in protocols.io, where a protocol can be assigned its own identifier (DOI) such that it can be cited independently in the future. For instructions see: http://journals.plos.org/plosone/s/submission-guidelines#loc-laboratory-protocols

We look forward to receiving your revised manuscript.

Kind regards,

Annalijn I Conklin, M.Sc., M.P.H., Ph.D.

Academic Editor

PLOS ONE

2. Please consider reporting of this observational study of clinical data using the STROBE checklist (http://www.strobe-statement.org) or the RECORD checklist (http://record-statement.org). Moreover, please ensure that no statements of causation are reported.

3. Please correct your reference to "p=0.000" to "p<0.001" or as similarly appropriate, as p values cannot equal zero.

5. We note you have included a table to which you do not refer in the text of your manuscript. Please ensure that you refer to Table 8-9 in your text; if accepted, production will need this reference to link the reader to the Table.

Reviewers' comments:

Reviewer's Responses to Questions

**Comments to the Author**

1. Is the manuscript technically sound, and do the data support the conclusions?

Reviewer #1: Partly

Reviewer #2: Yes

2. Has the statistical analysis been performed appropriately and rigorously? 

Reviewer #1: No

Reviewer #2: Yes

3. Have the authors made all data underlying the findings in their manuscript fully available?

Reviewer #1: Yes

Reviewer #2: Yes

4. Is the manuscript presented in an intelligible fashion and written in standard English?

Reviewer #1: Yes

Reviewer #2: Yes

5. Review Comments to the Author

Reviewer #1: The manuscript by Mondschein and colleagues aims to investigate gender bias in waiting times for selected health care services in Chilean public health system. The topic is interesting and general idea behind the study could attract the attention of Journal's readers; unfortunately, there are some serious flaws which prevent me from recommending the manuscript in its present form. Below you will find my comments and suggestions which, in my opinion, could improve your research and the way it is communicated.

1. First of all, I think you should use more sophisticated methods to disentangle relationships involved. I think that regression-based analyses would be more appropriate in this case. The use of multiple regression would allow to account for the impact of various factors at the same time and test interactions between independent variables. What is more, because your observations are clustered (particular patients, types of provider, insurance status) I suggest that you consider multilevel approach to regression. Your results would be more robust than with the use of t-test.

2. I think you should explicitly state that you investigate gender bias in your title; it would be more informative.

3. Each time you use abbreviations first time in the text, they should be defined; this applies also to abstract (see GES, AUGE...).

4. The abstract itself provides insufficient details on your study. You do not mention the method used, the timespan of the analysis, the characteristics of the dataset. These critical details should be explicitly provided here.

5. Both in abstract and conclusion you describe bias as 'unexpected'. Why are they unexpected? Several studies (you reference some of them) provide evidence on gender bias favoring men and thus your findings does not seem to be surprising.

6. The introductory part of your manuscript is a bit chaotic. The first and the last paragraphs of this part are concerned with Chilean health care and gender bias and between the two you review the previous literature on gender bias. I suggest that you reorganize it. Also, you start the introduction with explicit statement on the aim of your study. Usually, first some background information is provided, followed by brief literature review and this leads to formulation of knowledge gap. Here, some details which are typical for scientific introductions are missing, like the knowledge gap and explicit statement of your study's original contribution. Please reorganize this section according to more standard form.

7. Based on the referenced study [11] you state that '...healthcare coverage was twice as high for men as for women.' in Poland. According to the study [11] this statement applies to 'voluntary private insurance' which might be (and in fact it is here) very different than 'healthcare coverage'.

8. First two paragraphs of the 'Overview of study design' are not something which could be labeled as materials and methods. This rather belongs to the content you now have in your very first paragraph of the introduction.

9. More explanation is needed to describe why your approach selects only 14 of 80 diseases included in GES.

10. Lines 134-141 should rather be a footnote in the table than the main text which should describe findings instead.

11.More details on defining 'type of provider' variable would be beneficial.

12. Line 217: '... population in . ...'; it seems you have a typo.

13. Generally, the materials and methods and results sections would look different if you follow my comment on the use of regression-based analysis; therefore, I do not give more detailed recommendation on particular points therein.

14. I like your discussion and the way you interpret your results; however, again, more could be done to make this section more standard in terms of usual content of discussion sections in scientific writing. I suggest to include general picture of your main findings at the beginning of discussion. But what is more important and in fact critical for decent discussion, you should definitely compare your results to previous studies from Chile and other settings. Moreover, you do not discuss limitations of your study at all while you should do so.

To conclude, there are several flaws that require improvements, including the study design itself. Though, I find the potential of this study as high and encourage Authors to make suggested changes.

Reviewer #2: This is very interesting study and may have significant policy implications for addressing gender differences in access to healthcare services in the Chilean public health system. However, there are some short comings which need to be addressed before it is considered for publication.

Abstract: The abstract needs restructuring in the following order: background, objectives, methods, findings and conclusion along with policy implications. The abstract is lacking policy implications. For this purpose, a couple of sentences may be added to the last of abstract.

Introduction:Introduction section is well written. Sufficient literature is reviewed. However, the authors have mostly cited the gender-differences from the western world. From developing world, along with an Indian study already cited, cite some more studies, such as those from Panezai, S., Ahmad, M. M., & Saqib, S. E. (2017). Factors affecting access to primary health care services in Pakistan: a gender-based analysis. Development in practice, 27(6), 813-827 and Panezai, S., Ahmad, M. M., & e Saqib, S. (2020). A Gender-Based Assessment of Utilization of Primary Health Care Services and Associated Factors in Pakistan. Ponte Journal, 76(1/1). Citing these studies, will help compare gender-differences from both the developing and developed word perspectives.

Methods: Methods section is sufficiently explained. Designed elaborated and data described. The author(s) should give full names to the abbreviations such as FONASA and GES at first use.

Results: Data is well presented and sufficiently explained.

Discussion: The discussion section is lacking comparative analysis of current findings with those of existing studies. In this section, the data presented need to be compared with the existing literature and efforts should be put to find similarities and differences if any. The authors should try to convince the policy makers that that how meaningful are these differences? Lastly, this study has not mentioned clearly the potential limitations, the authors must document the limitations of the current study.

Conclusion: The conclusion section fails to stipulate policy implications of the current study. The authors should come up with concrete policy level suggestions for addressing the gender differences.

Overall: This manuscript has the potential to be published provided that the above mentioned issues are addressed.

6. PLOS authors have the option to publish the peer review history of their article (what does this mean?). If published, this will include your full peer review and any attached files.

Reviewer #1: Yes: Błażej Łyszczarz

Reviewer #2: Yes: Sanaullah Panezai, PhD

---

## [Author Response · Author response to Decision Letter 0]

2 Jun 2020

We thank the reviewers and associate editor for their input, which significantly improved the quality of this manuscript, and hope that our new improved version of the paper addresses and clarifies all their questions and comments.

Reviewer 1

1. First of all, I think you should use more sophisticated methods to disentangle relationships involved. I think that regression-based analyses would be more appropriate in this case. The use of multiple regression would allow to account for the impact of various factors at the same time and test interactions between independent variables. What is more, because your observations are clustered (particular patients, types of provider, insurance status) I suggest that you consider multilevel approach to regression. Your results would be more robust than with the use of t-test.

We thank your thoughtful comment and following your advice we have used a multilevel model, in addition to the Welch t-test. This substantial change in the paper, due to this new approach, is included in sections Materials and Methods, Results, and Discussion.

2. I think you should explicitly state that you investigate gender bias in your title; it would be more informative.

Following your suggestion, we have included the gender bias in the title.

3. Each time you use abbreviations first time in the text, they should be denied; this applies also to abstract (see GES, AUGE...).

Done it.

4. The abstract itself provides insufficient details on your study. You do not mention the method used, the timespan of the analysis, the characteristics of the dataset. These critical details should be explicitly provided here.

Following your advice, we have rewritten the abstract using the following format: background, objective, methods, results and conclusion. We hope the new structure provides clearer details on our study.

5. Both in abstract and conclusion you describe bias as 'unexpected'. Why are they unexpected? Several studies (you reference some of them) provide evidence on gender bias favoring men and thus your findings does not seem

to be surprising.

The GES plan was introduced in order to level up differences in access and opportunities for the Chilean population. This is why we were not expecting to find such significant differences in waiting time between women and men. We have rewritten the Abstract and Conclusions to clarify this observation.

6. The introductory part of your manuscript is a bit chaotic. The first and the last paragraphs of this part are concerned with Chilean health care and gender bias and between the two you review the previous literature on gender bias. I suggest that you reorganize it. Also, you start the introduction with explicit statement on the aim of your study. Usually, first some background information is provided, followed by brief literature review and this leads to formulation of knowledge gap. Here, some details which are typical for scientific introductions are missing, like the knowledge gap and explicit statement of your study's original contribution. Please reorganize this section according to more standard form.

Thank you for your comment. We have reorganized the Introduction according to your suggestion and believe it is much better now.

7. Based on the referenced study [11] you state that '...healthcare coverage was twice as high for men as for women.' in Poland. According to the study [11] this statement applies to 'voluntary private insurance' which

might be (and in fact it is here) very different than 'healthcare coverage'.

We have rewritten the citation. We apologize for this.

8. First two paragraphs of the 'Overview of study design' are not something which could be labeled as materials and methods. This rather belongs to the content you now have in your very first paragraph of the introduction.

We have reorganized the introduction, including the first paragraphs you were referring to.

9. More explanation is needed to describe why your approach selects only 14 of 80 diseases included in GES.

After your comment we review how waiting time was computed with great detail. We finally included 16 GES-OG in the analysis corresponding to 15 diseases. In subsection Data (within section Materials and Methods), we have included the following paragraph explaining how waiting time is computed:

“We construct the waiting time (WT) as the time lapse between two interactions that have an explicit OG. It is worth noticing that not all diseases have the same guarantees, both in terms of maximum time allowed and

also in the part of the care pathway that is covered. While for some diseases there are OGs for diagnostic confirmation, treatment and follow-up while, others include only treatment. Moreover, there are diseases in the GES plan for which the diagnostic confirmations do not generate an additional monetary transfer from FONASA to the provider that are not recorded in the database, preventing the construction of the WT (for instance the diagnostic confirmation of depression)."

10. Lines 134-141 should rather be a footnote in the table than the main text

which should describe findings instead.

We followed your suggestion and moved the columns' description to a footnote in the table. We have only included those that needed further explanation.

11. More details on defining 'type of provider' variable would be beneficial.

We included the following paragraph in Data subsection with more details: “High complexity providers are larger hospitals (more than 300 beds) with 20% of its bed capacity assigned to critical care. They provide access to all

(or most) subspecialties, diagnostic and treatment equipment. Medium complexity providers are hospitals with 31 to 300 beds, with a small proportion of beds for critical care, if any. They have lower level of infrastructure and equipment (no radiotherapy or chemotherapy, for instance) and they do not have all the subspecialties staffed. Low complexity providers are small hospitals with up to 30 beds, delivering only basic care. Health centers and health reference centers are medium-complexity institutions that provide only outpatient health services. However, they are responsible for referring patients within their health area who need care exceeding the center's level of complexity. In such cases, the institution pays for treatment and receives the FONASA reimbursement."

12. Line 217: '... population in . ...'; it seems you have a typo.

This sentence is not longer part of the manuscript.

13. Generally, the materials and methods and results sections would look different if you follow my comment on the use of regression-based analysis; therefore, I do not give more detailed recommendation on particular points

therein.

Materials and Methods, Results and Discussion sections have substantially changed.

14. I like your discussion and the way you interpret your results; however, again, more could be done to make this section more standard in terms of usual content of discussion sections in scientific writing. I suggest to include general picture of your main findings at the beginning of discussion.

But what is more important and in fact critical for decent discussion, you should definitely compare your results to previous studies from Chile and other settings. Moreover, you do not discuss limitations of your study at all while you should do so.

We rewrote the Discussion section following your recommendations. We also mentioned the novelty of our study compared with the current literature on the subject. We included the following paragraph for limitations:

“The main limitation of this study is that we did not explicitly consider the congestion of the healthcare system. Including the latter would be useful to understand compliance rates and its possible impact in WT. Further research is needed to understand if under the pressure of congestion, gender bias is exacerbated.

Reviewer 2

1. Abstract: The abstract needs restructuring in the following order: background, objectives, methods, findings and conclusion along with policy implications. The abstract is lacking policy implications. For this purpose, a couple of sentences may be added to the last of abstract.

Following your suggestion, we have restructured the abstract accordingly.

2. Introduction: Introduction section is well written. Sufficient literature is reviewed. However, the authors have mostly cited the gender-differences from the western world. From developing world, along with an Indian study already cited, cite some more studies, such as those from Panezai, S., Ahmad, M. M., & Saqib, S. E. (2017). Factors affecting access to primary health care services in Pakistan: a gender-based analysis. Development in practice, 27(6), 813-827 and Panezai, S., Ahmad, M. M., & e Saqib, S. (2020). A Gender-Based Assessment of Utilization of Primary Health Care Services and Associated Factors in Pakistan. Ponte Journal, 76(1/1). Citing these studies, will help compare gender-differences from both the developing and developed word perspectives.

We agree with you that the paper by Panezai et.al (2020) is relevant to our work, and it has been included as follows: \\In [12] the authors study gender-based utilization factors of primary health centers in Pakistan, finding statistical differences in predisposing, enabling and need factors."

[12] Panezai, S., Ahmad, M. M., & e Saqib, S. (2020). A Gender-Based Assessment of Utilization of Primary Health Care Services and Associated Factors in Pakistan. Ponte Journal, 76(1/1).

3. Methods: Methods section is sufficiently explained. Designed elaborated and data described. The author(s) should give full names to the abbreviations such as FONASA and GES at first use.

Following your suggestion, we have included the full names for both FONASA and GES upon their first use.

4. Results: Data is well presented and sufficiently explained.

Thank you very much. We notice that in the new manuscript, this section has been upgraded to include the results of the multilevel regression model.

5. Discussion: The discussion section is lacking comparative analysis of current findings with those of existing studies. In this section, the data presented need to be compared with the existing literature and efforts should

be put to find similarities and differences if any. The authors should try to convince the policy makers that that how meaningful are these differences? Lastly, this study has not mentioned clearly the potential limitations, the

authors must document the limitations of the current study.

Materials and Methods, Results, and Discussion sections have substantially changed in the new version of the manuscript. We hope this new version is more complete.

We included the following paragraph for limitations: “The main limitation of this study is that we did not explicitly consider the congestion of the healthcare system. Including the latter would be useful to understand compliance rates and its possible impact in WT. Further research is needed to understand if under the pressure of congestion, gender bias is exacerbated".

6. Conclusion: The conclusion section fails to stipulate policy implications of the current study. The authors should come up with concrete policy level suggestions for addressing the gender differences.

We have enhanced the conclusion including more concrete policy suggestions:

“Our analysis shows that the existence of explicit opportunity guarantees for GES--OG does not prevent bias when considering the timeliness of treatment between women and men. This bias is impacted by the difference among providers and health districts, along with other observable patients' characteristics such as age and insurance holder status. We believe, with the limited evidence at hand, that differences in waiting times are most likely a product of a complex combination of several factors, where the role of women in society might be a fundamental component. Understanding these factors is part of our ongoing research. Once the reasons behind these biases are known, more specific, differentiated, gender-oriented policies should be implemented. In the meantime, positive

actions that facilitate timely treatment for women should be considered, especially for those between 40 and 54 years old."

---

## [Decision Letter · Decision Letter 1]

16 Jul 2020

PONE-D-20-03609R1

Gender Bias in the Chilean public health system: Do we all wait the same?

PLOS ONE

Dear Dr. mondschein,

Thank you for submitting your manuscript to PLOS ONE. After careful consideration, we feel that it has merit but does not fully meet PLOS ONE’s publication criteria as it currently stands. Therefore, we invite you to submit a revised version of the manuscript that addresses the points raised during the review process.

Please edit the exhibits so that each one can be understood on its own; explain abbreviations and what key values mean; use clear titles as suggested by the Reviewer as I agree with this perspective.

We look forward to receiving your revised manuscript.

Kind regards,

Annalijn I Conklin, M.Sc., M.P.H., Ph.D.

Academic Editor

PLOS ONE

Additional Editor Comments (if provided):

Please edit the exhibits so that each one can be understood on its own; explain abbreviations and what key values mean; use clear titles as suggested by the Reviewer.

Reviewers' comments:

Reviewer's Responses to Questions

**Comments to the Author**

1. If the authors have adequately addressed your comments raised in a previous round of review and you feel that this manuscript is now acceptable for publication, you may indicate that here to bypass the “Comments to the Author” section, enter your conflict of interest statement in the “Confidential to Editor” section, and submit your "Accept" recommendation.

Reviewer #1: (No Response)

Reviewer #2: All comments have been addressed

2. Is the manuscript technically sound, and do the data support the conclusions?

Reviewer #1: Yes

Reviewer #2: Yes

3. Has the statistical analysis been performed appropriately and rigorously? 

Reviewer #1: Yes

Reviewer #2: N/A

4. Have the authors made all data underlying the findings in their manuscript fully available?

Reviewer #1: Yes

Reviewer #2: Yes

5. Is the manuscript presented in an intelligible fashion and written in standard English?

Reviewer #1: Yes

Reviewer #2: Yes

6. Review Comments to the Author

Reviewer #1: Dear Authors,

congratulations, you have made a great improvement in this revised manuscript. I really enjoyed reading it and appriciate what you have done.

Let me just point to one more issue. I think you could do more to make your tables and figures more appealing. I mean, they could be more self-standing and comprehensible without referring to text. Therefore, I think you could explain all the abbreviations also in tables'/figures' 'Notes:' (at least at the first time they are used in tables/figures), see for example 'GES' in T1, 'GES-OG' in T2. Also, in my opinion, the abbreviations in the titles are not recommended; your title in T5 could be much more informative and give more details on what the numbers below show. For tables 6 and 7, again in the 'Notes:' under the tables I think you could provide a brief explanation of how to interpret the numbers; what does negative/positive value mean. In figure 1, you do not have axis labels. In figure 2, the term 'Age group [5n,5n+4]' is not clear for me.

Generally, try to think about presenting tables and figures in the way that allows to find out as much as possible from tables and figures solely, with not much reference to the text. Notes under the tables/figures might be very useful for this purpose.

Reviewer #2: The authors have done good job and have incorporated all the suggestions of reviewer. This manuscript should be accepted for publication now.

7. PLOS authors have the option to publish the peer review history of their article (what does this mean?). If published, this will include your full peer review and any attached files.

Reviewer #1: **Yes: **Błażej Łyszczarz

Reviewer #2: No

---

## [Author Response · Author response to Decision Letter 1]

11 Aug 2020

Dear Academic Editor and Reviewers

Thank you very much for the time you have devoted to reading our paper "Bias in the Chilean public health system: Do we all wait the same?" (PONE-D-20-03609). In what follows, we answer in detail the reviewer's comments.

Reviewer 1

1. Dear Authors, congratulations, you have made a great improvement in this revised manuscript. I really enjoyed reading it and appreciate what you have done.

We appreciate your positive feedback and the insightful comments you have provided in your reports. Our paper became a better one because of that.

2. Let me just point to one more issue. I think you could do more to make your tables and figures more appealing. I mean, they could be more self-standing and comprehensible without referring to text.

Following your advice, we have improved the titles of tables and included notes to make them more self-standing. 

3. Therefore, I think you could explain all the abbreviations also in tables'/figures' 'Notes:' (at least at the first time they are used in tables/figures), see for example 'GES' in T1, 'GES-OG' in T2. Also, in my opinion, the abbreviations in the titles are not recommended; your title in T5 could be much more informative and give more details on what the numbers below show. For Tables 6 and 7, again in the 'Notes:' under the tables I think you could provide a brief explanation of how to interpret the numbers; what does negative/positive value mean. In figure 1, you do not have axis labels. In figure 2, the term 'Age group [5n,5n+4]' is not clear for me.

Following your suggestion, we have eliminated all abbreviations from the titles of tables and figures and added the explanation of any abbreviation at the bottom of figures/tables. We have changed the title of Table 5, including further details on what the numbers show. For Tables 6 and 7 we have included a note with a brief interpretation of the main findings. We have labeled the axis of Figure 1, and we have also improved the label of the group age in Figure 2. 

4. Generally, try to think about presenting tables and figures in the way that allows to find out as much as possible from tables and figures solely, with not much reference to the text. Notes under the tables/figures might be very useful for this purpose.

Tables and figures have been improved, making them easier to understand without referring to the main text. 

Reviewer 2

1. The authors have done good job and have incorporated all the suggestions of reviewer. This manuscript should be accepted for publication now.

We thank the reviewer for the positive feedback.

We thank the reviewers and associate editor for their input, which significantly improved the quality of this manuscript, and hope that this revised version of the paper addresses all their comments.

---

## [Editor Report · Decision Letter 2]

7 Sep 2020

Gender Bias in the Chilean public health system: Do we all wait the same?

PONE-D-20-03609R2

Dear Dr. mondschein,

We’re pleased to inform you that your manuscript has been judged scientifically suitable for publication and will be formally accepted for publication once it meets all outstanding technical requirements.

Kind regards,

Annalijn I Conklin, M.Sc., M.P.H., Ph.D.

Academic Editor

PLOS ONE

Additional Editor Comments (optional):

Thank you for making improvements to your exhibits, and all suggested edits.
---

## [Editor Report · Acceptance letter]

15 Sep 2020

PONE-D-20-03609R2 

Gender Bias in the Chilean public health system: Do we all wait the same?  

Dear Dr. Mondschein:

I'm pleased to inform you that your manuscript has been deemed suitable for publication in PLOS ONE. Congratulations! Your manuscript is now with our production department. 

Kind regards, 

on behalf of

Dr. Annalijn I Conklin 

Academic Editor

PLOS ONE